# V-Porphyrins Encapsulated or Supported on Siliceous Materials: Synthesis, Characterization, and Photoelectrochemical Properties

**DOI:** 10.3390/ma15217473

**Published:** 2022-10-25

**Authors:** Zhannur K. Myltykbayeva, Anar Seysembekova, Beatriz M. Moreno, Rita Sánchez-Tovar, Ramón M. Fernández-Domene, Alejandro Vidal-Moya, Benjamín Solsona, José M. López Nieto

**Affiliations:** 1Instituto de Tecnología Química, Universitat Politècnica de València-Consejo Superior de Investigaciones Científicas, Avenida de los Naranjos s/n, 46022 Valencia, Spain; 2Al-Farabi Kazakh National University, 71 Al-Farabi Ave., Almaty 050040, Kazakhstan; 3Departament d’Enginyeria Química, Universitat de València, Av. de les Universitats, s/n, 46100 Burjassot, Spain

**Keywords:** vanadyl porphyrin, mesoporous silica, SBA-15, characterization, supported, electrochemical characterization, photoelectrocatalysis

## Abstract

Metalloporphyrin-containing mesoporous materials, named VTPP@SBA, were prepared via a simple anchoring of vanadyl porphyrin (5,10,15,20-Tetraphenyl-21*H*,23*H*-porphine vanadium(IV) oxide) through a SBA-15-type mesoporous material. For comparison, vanadyl porphyrin was also impregnated on SiO_2_ (VTPP/SiO_2_). The characterization results of catalysts by XRD, FTIR, DR-UV-vis, and EPR confirm the incorporation of vanadyl porphyrin within the mesoporous SBA-15. These catalysts have also been studied using electrochemical and photoelectrochemical methods. Impedance measurements confirmed that supporting the porphyrin in silica improved the electrical conductivity of samples. In fact, when using mesoporous silica, current densities associated with oxidation/reduction processes appreciably increased, implying an enhancement in charge transfer processes and, therefore, in electrochemical performance. All samples presented *n*-type semiconductivity and provided an interesting photoelectrocatalytic response upon illumination, especially silica-supported porphyrins. This is the first time that V-porphyrin-derived materials have been tested for photoelectrochemical applications, showing good potential for this use.

## 1. Introduction

Porphyrin and metalloporphyrin complexes present peculiar physicochemical characteristics and are easily tunable, which offers the possibility of being used in a good number of technological applications in fields such as biomedical applications, solar cells, chemical and biochemical sensors, and nanowires/nanomotors [1,2,3,4,5]. Due to their stable macrocyclic structure, porphyrins constitute an exceptional class of ligands with tailorable chemical properties, which is of high interest in the development of selective homogeneous or heterogeneous catalysts and photocatalysts [1,6,7,8,9,10,11,12,13]. In fact, porphyrin and metalloporphyrin complexes have been used pure [6,7,8,9], supported on several materials [10,11,12,13], immobilized on silica or mesoporous materials [13,14,15,16,17,18,19,20], or incorporated into MOFs [21,22,23], presenting a high ability to catalyze a variety of reactions such as oxidations, epoxidations, hydrogenation, or deoxygenation.

The oil currently extracted is, on average, heavier than that obtained several decades ago. This oil, apart from presenting larger hydrocarbons, typically contains larger concentrations of metals, sulfur, oxygen, and nitrogen. Thus, V is one of the most abundant metals in the oil and is commonly present in porphyrin structures. Therefore, the search for new applications of these V-containing petroporphyrins is highly interesting. Vanadium(IV)-porphyrin-containing materials are also paramount [24,25,26,27], in which special attention has been paid to the use of Vanadyl-porphyrins incorporated on the surface of supports or encapsulated in mesoporous materials [13,28,29,30,31,32]. It is well-known that changes in the type of functional groups and/or in the position of the substituents in the porphyrin macrocycle can give rise to modifications in the physicochemical and catalytic properties [6,7,8,9], but significant changes in physicochemical characteristics could also occur when incorporating metalloporphyrins into inorganic oxides [10,11,12,13,14,15,16,17,18,19,20,28,29,30,31,32]. The role of vanadium is important in these applications. Further, it has been reported that vanadium in other surroundings, as in V-doped nickel sulfide nanoflowers, exhibits a notable electrochemical performance [33].

Vanadyl-porphyrins present interesting electrical properties [13,24,25,26,27] and could be of interest to be used as catalysts in electrochemical reactions. However, to the best of our understanding, no research article has been published that focused on the photoelectrochemical characteristics of bulk and supported vanadyl-porphyrins. Although not with V, other porphyrin-based materials have been shown to exhibit some photoelectrochemical activity [34,35,36,37,38]. Photoelectrochemistry studies, in general, processes taking place at the interface between a semiconducting electrode and an electrolyte in which, under the influence of illumination with suitable energy, electron–hole pairs are generated. Photoelectrochemistry has numerous attractive applications [39,40,41,42,43,44], such as the split of a water molecule to obtain its fundamental constituents (H_2_ and O_2_), the advanced oxidation of persistent and toxic organic pollutants in water treatment processes, the photoelectrochemical reduction of greenhouse gases such as CO_2_, or the fabrication of dye-sensitized solar cells (DSSC). Semiconductor metal oxides, such as TiO_2_, WO_3_, or different copper oxides, are increasingly studied as photoelectrodes, due to their different characteristics.

In the present work, V-porphyrin-based catalysts have been synthesized. Thus, V-porphyrin encapsulated in SBA-15 has been prepared, using 5,10,15,20-Tetraphenyl-21*H*,23*H*-porphine vanadium(IV) oxide (VTPP) as the V-source. For comparison, pure VTPP and VTPP supported on amorphous SiO_2_ have been also prepared, characterized, and tested in photoelectrochemical applications.

## 2. Materials and Methods

### 2.1. Materials

5,10,15,20-Tetraphenyl-21*H*,23*H*-porphine vanadium(IV) oxide (VTPP), C_44_H_28_N_4_OV, was purchased from Aldrich and used without further purification. Pluronic (P123, Sigma Aldrich, St. Louis, MO, USA), hydrochloric acid (HCl, 37%; Fisher Chemical, Hampton, NH, USA), butanol (Sigma Aldrich), and tetraethyl orthosilicate (Si(OC_2_H_5_)_4_, TEOS, Sigma Aldrich) were used in the synthesis of mesoporous materials. In addition, ammonium niobate (V) oxalate hydrate (C_4_H_4_NNbO_9_·xH_2_O, Sigma Aldrich) was also used in the preparation of Nb-containing mesoporous material.

### 2.2. Synthesis of Catalysts and Supports

#### 2.2.1. Synthesis of Mesoporous Pure Silica and Nb-Doped SBA-15

Nb-free and Nb-containing mesoporous materials of SBA-15 structure were synthesized according to the modified procedure previously reported in [44], which is summarized in Appendix A. In a typical synthesis using the nonionic triblock copolymer surfactant Pluronic P123 (EO_20_PO_70_EO_20_, MW = 5800, from Aldrich), 4.65 g of Pluronic P123 was dissolved in 167 mL of water (milli Q) with 9.2 g of HCl 37% and 4.65 g of n-butanol, and the mixture was stirred for 1 h. Next, 10 g of tetraethyl orthosilicate (TEOS) was added, and the resulting white liquid was kept under stirring for 24 h. Then, the solution was transferred to a Teflon-coated stainless-steel autoclave, and the synthesis took place under static hydrothermal conditions at 100 °C for 24 h. The solid formed was filtered and washed with 2 L of deionized water, drying it at 60 °C for 12 h. This will be named SBA.

In the case of Nb-doped SBA-15, named SBA(Nb), a 2 wt% of Nb (using ammonium niobate (V) oxalate hydrate as the Nb-precursor) was co-added, with the remaining TEOS quantity following the same synthesis procedure as that used for the preparation of SBA-15.

Finally, the samples were calcined in order to eliminate the remaining organic template. In all cases, the samples were heat-treated at 550 °C for 4 h (1.5 °C/min ramp) in an open-air muffle.

#### 2.2.2. Preparation of VTPP Encapsulated in Mesoporous Material

Encapsulation of the VPTT requires a three-necked balloon, a silicone bath, a heating plate, a cooling column, and a vacuum pump (or rotary evaporator). The VPTT solution was obtained by stirring for 6 h (until total solubility) a methanol solution containing VTPP. On the other hand, 1 g of calcined support, i.e., mesoporous materials (SBA-15, or Nb-containing SBA-15), was treated for 30 min under vacuum to remove the air/H_2_O from the mesopore channels. After this, the VPTT-containing methanol solution was incorporated into the mesoporous material and placed at 65 °C for 6 h under reflux to prevent the methanol from evaporating. Finally, the mixture was filtered, washed with methanol (to remove weakly bound VTPP), and dried in an oven at 60 °C overnight (Appendix A). The catalysts are named xVP@SBA or xVP@SBA(Nb), in which x is the VTPP content in the catalysts, i.e., 5 or 20 wt% of VTPP.

#### 2.2.3. Preparation of SiO_2_-Supported VTPP Catalysts

SiO_2_-supported VTPP catalysts have been prepared following the same procedure as that of xVTPP@SBA. In this case, SiO_2_ nanopowder (surface area of 225 m^2^ g^−1^) (Aldrich) was used as a support, using a ratio of 200 mg of VTPP for 1g of SiO_2_ (i.e., 20 wt% of VTPP). The catalyst is named 20VP/SiO_2_.

### 2.3. Characterization of Catalysts and Supports

The specific surface area of catalysts and supports has been determined by the BET method from N_2_ adsorption isotherms at −196 °C, measured in a Micromeritics TriStar 3000 instrument. X-ray diffraction (XRD) patterns of powder solids were collected with a PANalytical CUBIX instrument equipped with a graphite monochromator, employing Cu Kα radiation (λ = 0.1542 mm) and operated at 45 kV and 4 mA. X-ray diffraction (XRD) patterns of solids were collected with a PANalytical CUBIX Instrument (Malvern, UK). Quantitative EDX analysis was carried out using an Oxford LINK ISIS System with the SEMQUANT program coupled to a JEOL 6300 microscope operating at 20 Kv (Tokyo, Japan), whereby scanning electron microscopy (SEM) images were collected.

IR spectra were measured with a Nicolet 205xB spectrophotometer (at a spectral resolution of 1 cm^−1^ and 128 accumulations per scan) in the 450–4000 cm^−1^ region, (Madison, WI, USA). Dried catalysts (ca. 20 mg) were mixed with 100 mg of dry KBr and pressed into a disk (600 kg cm^−2^). Diffuse reflectance UV-vis (DR-UV-vis) spectra were recorded within the 200–800 nm range using a Varian spectrometer model Cary 5000 (Palo Alto, CA, USA). In the case of pure VTPP, this was studied as received or diluted in CH_2_Cl_2_. EPR spectra were obtained at −170 °C using an EMX-12 Bruker spectrometer (Billerica, MA, USA) and working at the X band, with a frequency modulation of 100 kHz and 1 G amplitude.

### 2.4. Electrochemical Measurements

The electrochemical study of the catalysts was carried out using different techniques: electrochemical impedance spectroscopy (EIS) measurements, Mott–Schottky (MS) analysis, and cyclic voltammetries. Additionally, a photoelectrochemical characterization was performed by polarizing the samples under dark–light cycles.

All these experiments were performed in a three-electrode configuration cell with a 0.1 M Na_2_SO_4_ aqueous solution as the electrolyte, except for the cyclic voltammetries, for which the electrolyte was 10 mM Fe(CN)_6_K_4_ + 0.1 M Na_2_SO_4_. The materials were deposited on an FTO to conduct the measurements, and 0.5 cm^2^ of area was exposed to the electrolyte. A platinum wire was used as the counter electrode, and an Ag/AgCl (3M KCl) electrode was used as the reference electrode. The electrodes were immersed in the electrolyte and connected to a potentiostat.

EIS experiments were carried out at 0.3 V (vs. Ag/AgCl) over a frequency range from 100 kHz to 10 mHz with a signal amplitude of 0.01 V. Mott–Schottky analyses were carried out by sweeping the potential from 0.5 to −0.5 V, with an amplitude signal of 0.01 V at a frequency value of 5 kHz. Cyclic voltammetries were performed in the range potential between −0.1 V and 0.6 V at 10 mV s^−1^.

Photocurrent density as a function of the applied potential was registered by chopped light radiation (16 s dark/4 s light) with a potential scan from 0.24 V to 1.02 V. For the photoelectrochemical tests, a wavelength of 365 nm was used.

## 3. Results and Discussion

### 3.1. Characterization of Catalysts

It is well-known that the pore size of mesoporous SBA-15 can be tuned from 3.8 nm to 18.6 nm [45,46], which permitted us to encapsulate the VTPP. Appendix A shows the structure of both the mesoporous SBA-15 material and VTPP, whereas Table 1 shows the main characteristics of the samples prepared: (i) pure V-porphyrin, (ii) V-porphyrin encapsulated on a mesoporous SBA-15 silica with two different VTTP loadings of 5 and 20 wt%, (iii) V-porphyrin supported on amorphous silica with two different VTTP loadings of 5 and 20 wt%, and (iv) V-porphyrin (20 wt%) on a Nb-modified SBA-15 matrix.

As can be seen in Table 1, the surface area of the encapsulated samples on SBA-15 are high, exceeding 400 m^2^ g^−1^ even with high VTTP loadings, although remarkably lower than that of the pure support (>700 m^2^ g^−1^). In the case of the silica-supported samples, surface areas lower than 200 m^2^ g^−1^ were attained in accordance with the surface area of the silica support.

Figure 1A shows the small-angle X-ray scattering of mesoporous supports and the corresponding VTPP-containing catalysts. All of them, supports and catalysts, show characteristic reflections of two-dimensional mesoporous structures with a SBA-15 structure (spatial group P6mm) and a pore size of ca. 8 nm [46], i.e., reflections with (100), (110), and (200) Miller indices, which are shown at ca. 0.96, 1.66, and 1.91°, respectively, for pure SBA-15. Only a slight shift for Nb-containing SBA-15 Nb is observed.

On the other hand, the XRD patterns of VTPP-containing materials are shown in Figure 1B. We must indicate that a broad band at ca. 23.49° related to the pseudo-crystallinity of these materials is only observed in the case of the supports, i.e., SBA and SBA(Nb) (Figure 1B, patterns a and b). However, in the case of VTPP-containing mesoporous materials (Figure 1, patterns c to e), the XRD patterns show several reflections at 2θ = 8.80; 12.28; 20.56; 31.83; 44.54°, which indicate the presence of VTPP with a polycrystalline structure [47,48], as observed for pure VTPP (see Appendix A, pattern a), related to a tetragonal phase structure with space P4_3_ [48]. We must relate that the SiO_2_-supported VTPP catalyst (Appendix A, pattern b) shows also a similar XRD pattern as those observed for SBA-15-based catalysts, suggesting that in any case, the VTPP compound was not modified during the catalyst preparation step.

The surface areas of mesoporous supports, after calcination at 550 °C, are shown in Table 1. The mesoporous supports present a surface area higher than 700 m^2^ g^−1^, whereas those related to SiO_2_ present a surface area of ca. 200 m^2^ g^−1^. The nitrogen adsorption-desorption isotherms of mesoporous supports are shown in Appendix A. A type-IV adsorption isotherm with a type-H1 hysteresis loop, which is characteristic of 1D cylindrical regular pores and capillary condensation [45,49], was observed in the isotherms of both supports and catalysts.

The FT-IR spectra of VTPP-containing samples in the 4000–450 cm^−1^ range are comparatively presented in Figure 2A, and those in the 3750–2750 cm^−1^ range are shown in Figure 2B. For comparison, the FT-IR spectra of pure VTPP and some catalysts and supports are shown in Appendix A.

In the case of infrared spectra of supports, i.e., SBA-15 (Figure 2A, spectrum a), a broad band at ca. 3472 cm^−1^ can be seen, which is related to molecular water hydrogen bonded to Si-OH groups, as well as a band at ca. 1620 cm^−1^, related to bending vibrations of O-H bonds in OH groups [45,49]. In addition, the band at 1080 cm^−1^ (with a shoulder at 1180) could be due to asymmetric stretch vibrations of Si-O-Si, whereas the bands in 980–950 cm^−1^ could be related to Si-O in-plane stretching vibrations of the silanol (Si-OH) groups generated by the presence of defect sites, and the band at 795–790 cm^−1^ is related to symmetrical stretching vibrations of Si-O-Si bonds.

On the other hand, a comparison of spectra in the 2000–500 cm^−1^ region (Figure 2A and Appendix A) indicates the presence of bands at 1461, 1440, 1375, 1315, 1268, 1151, 1058, 1013, 1000, 983, 959, 837, 742, 712, and 698 cm^−1^ in the pure VTPP sample (Appendix A).

The presence of C–H groups was indicated by the bands at 2869 (CH_3_ asymmetric stretching), 2930 (symmetrical CH_2_), and 2967 cm^−1^ (asymmetrical CH_2_), in addition to those at 1443, 1373, and 705 cm^−1^ [14,25,50,51], whereas the presence of C-C groups was indicated by the band at 1465 cm^−1^ [51]. On the other hand, the presence of a pyrrole C-N bond was indicated by the band at 775 cm^−1^ [51], and the bands at 1058 and 844 are related to N-H vibration [50]. All these bands are also observed for VTPP-containing catalysts, although bands at 1268, 1151, 1058, 1013, 1000, 983, and 959 cm^−1^ appeared in the form of shoulders (because they overlap with the broad bands of Si-O-Si and Si-O-C) [14]. Accordingly, the presence of all these bands confirms the incorporation of VTPP in the catalysts, which is in agreement with previous results for V-porphyrins [18,26,27,28,29]. In addition, this suggests also that the VTPP is structurally unchanged and uniformly distributed in the mesoporous material [30].

The catalysts have been also characterized by DR-UV–vis spectroscopy, since most porphyrins present visible bands between 400 and 600 nm, usually called the Soret α and β bands [26,27,28,29,52,53]. The DR-UV–vis spectra of catalysts are shown in Figure 3, whereas those related to the pure VTPP sample are presented in Appendix A. DR-UV-vis spectra of catalysts show three bands: an intense band at 410 nm (Soret band) and a doublet at 533 nm (β band) and 572 nm (α band) that is related to Q bands, which is in good agreement with previous results for pure [24,25,26,27,28,29,43] or supported/encapsulated [30] vanadyl porphyrins.

In the case of pure VTPP samples (solid as received), the DR-UV-vis spectrum presents several bands, the most representative being at 353, 525, and 562 nm, in addition to a band at 304 nm. However, the DR-UV-vis spectrum of pure VTPP diluted in CH_2_Cl_2_ (3 mg VPTT in 30g CH_2_Cl_2_) presents characteristic bands at 407, 535, and 570 nm (Appendix A), which are in good agreement with those previously reported for V-porphyrins diluted in THF or dichloromethane [14,26,27,31,54,55,56].

In the same way, when comparing SiO_2_-supported catalysts with different VTPP contents (Appendix A), it can be seen that the intensity of the bands at 407 (I_407_) and at 574 nm (I_574_) changes depending on the concentration of VTPP in a solid state. Thus, the higher the concentration (or the lower the dispersion) of VTPP, the lower the I_407_/I_574_ ratio is. On the other hand, we must indicate that the position and intensity of the band at ca. 405 nm can be modified depending on the concentration or the pH in which the VTPP is present. Thus, it has been reported that 5,10,15,20-tetrakis(3,4-dimethoxyphenyl)-21*H*,23*H*-porphyrin entrapped in silica matrices can present a band at 425 or 465, depending on the pH of the mixture [14].

X-band spectra of VTPP-containing materials are shown in Figure 4, and the quantification of V(IV) content by EPR is presented in Table 1. For comparison, Appendix A shows the EPR spectrum of pure VTPP. All of these exhibited typical EPR spectra with neatly resolved ^51^V hyperfine lines at 120 K. They consist of eight broad lines separated by about 175 gauss that can be ascribed to the eight-line hyperfine structure from the ^51^V (I = 7/2) nucleus [57], without significant differences with the spectrum of pure VTPP (Figure 4, spectrum d). Figure 4B shows a detailed spectrum of sample 20VP@SBA, with a g_║_ and g_┴_ value of 1.962 and 1.984, respectively, and A_║_ and A_┴_ values of 176.6 gauss and 60.1 gauss, respectively. These parameters have been similar in all supported catalysts.

However, appreciable differences in the amount of V(IV) have been observed as a consequence of the VTPP content in each sample. These results suggest that, although the samples were washed with methanol during the preparation step, the possible elimination of VTPP during the washing step is negligible or very low. In addition, the amount of V(IV) determined by EPR (related to VTPP content) is in good agreement with the DR-UV-vis results. In addition, this information suggests that the encapsulation of the vanadyl porphyrin in the mesoporous SBA-15 is effectively achieved, and this encapsulation occurs without significant disruption in supramolecular organization [58]. In fact, FTIR, DR-UV-vis, and EPR spectra indicate no changes related to VTPP molecules when compared with pure VTPP and VTPP-containing siliceous catalysts.

### 3.2. Electrochemical Results

These VPP-based samples have been thoroughly characterized by electrochemical techniques. Thus, Figure 5 shows the Nyquist plot for VTPP-containing catalysts, whereas Figure 6 shows the Bode-module and Bode-phase plots for the different catalysts. According to Nyquist and the Bode-module plots (Figure 5 and Figure 6), the highest resistance was obtained for the pure VTPP porphyrin, whereas samples with 20% V-porphyrin supported on amorphous and mesoporous silica (i.e., 20VP/SiO_2_ and 20VP/SBA, respectively) provided the lowest impedance values.

The other two samples, namely, the one with 5% V-porphyrin in mesoporous silica (5VP@SBA) and the sample with 20% porphyrin in mesoporous Nb-doped SBA15 (20VP@SBA-Nb), gave intermediate impedance values. Accordingly, these results indicate, first, that supporting the porphyrin in silica improves the electrical conductivity of the samples, regardless of the type of the siliceous material used. On the other hand, it seems that reducing the amount of porphyrin from 20% to 5%, as well as adding niobium, resulted in somewhat lower electrical conductivities.

Figure 6 shows both the Bode-module (Figure 6A) and Bode-phase (Figure 6B) plots for the different VTTP-containing catalysts. Substantial differences are appreciated among the samples tested. Thus, the Bode-phase plot (Figure 6B) shows a clearly capacitive behaviour for the VTPP sample, with phase angles close to 90° for a large range of intermediate frequencies. The addition of both amorphous and mesoporous silica modifies the shape of the plots, and two time constants were discerned, especially for the samples with 20% porphyrin in mesoporous silica, i.e., 20VP@SBA and 20VP@SBA(Nb).

This change can be related to the higher surface area of these samples. In fact, as explained above, amorphous SiO_2_ provided a surface area of ca. 200 m^2^ g^−1^, whereas mesoporous silica had ca. 750 m^2^ g^−1^. The sample containing 5 wt% of VTPP in mesoporous silica also shows this feature in the Bode-phase representation, but the shape of the plot is not as evident, probably because the amount of porphyrin is not enough to fully cover the entire surface of the support.

In order to investigate whether the different porphyrin samples presented semiconducting behaviour, the capacitance of the interface between the samples (electrodes) and the electrolyte was measured as a function of the applied potential. Mott–Schottky (MS) analysis was performed by plotting the reciprocal square interfacial capacitance versus the applied potential. Figure 7 shows the MS plots for the different samples. First of all, it can be observed that a positive slope was obtained in all plots, according to the following equation for n-type semiconductors:(1)$$\frac{1}{C^{2}}\;=\;\frac{1}{C_{H}^{2}}\;+\;\frac{2}{\varepsilon_{r}\varepsilon_{0}eN_{D}}\left(E\;-\;E_{FB}\;-\;\frac{kT}{e}\right)$$
where *C* is the measured interfacial capacitance, *C_H_* is the capacitance of the Helmholtz layer, *ε_r_* is the relative permittivity or dielectric constant of the sample, *ε*_0_ is the vacuum permittivity (8.85 × 10^−14^ F/cm), *e* is the electron charge (1.60 × 10^−19^ C), *N_D_* is the density of defects (donor species for n-type semiconductors), *E* is the applied potential, *E_FB_* is the flat-band potential, *k* is the Boltzmann constant (1.38 × 10^−23^ J/K), and *T* is the absolute temperature.

Hence, for an n-type semiconductor, the plot 1/*C*^2^ vs. *E* yields a linear region with a positive slope, whose value is inversely proportional to the density of the donor species (*N_D_*). From Figure 7, it is clear, therefore, that all the samples presented n-type semiconducting behaviour. The value of the slope was similar for all the samples, although for VTPP, the slope was somewhat higher, indicating a lower density of defects. Samples with the vanadium porphyrin and dispersed in mesoporous silica (20VP@SBA and 5VP@SBA) showed slightly lower slope values, meaning that these samples had a higher donor density, although the values depend on the dielectric constant values of the different samples.

Figure 8 shows the cyclic voltammetries of the different samples. Two different peaks can be observed, one at around 0.17 V and the other at 0.3 V, which correspond, respectively, to the reduction and oxidation of the ferro/ferri electrolyte. The lowest current densities, both cathodic (reduction) and anodic (oxidation), were recorded for the VTPP and 20VP/SiO_2_ samples. Indeed, current density values for the VTPP sample were −0.3 mA/cm^2^ and 0.49 mA/cm^2^ (cathodic and anodic peaks, respectively), whereas for the 20VP/SiO_2_, these values were −0.31 mA/cm^2^ and 0.54 mA/cm^2^. When mesoporous silica was used, current densities notoriously increased, especially the oxidation current density for the 20VP@SBA samples, reaching 0.65 mA/cm^2^. With lower amounts of porphyrin (5VP@SBA), the anodic current density was slightly lower (0.61 mA/cm^2^) than for 20VP@SBA, indicating lower electrochemical activity towards oxidation processes. With the addition of Nb (20VP@SBA(Nb)), the anodic current density decreased to 0.57 mA/cm^−2^, and so, in this case, the presence of Nb in the porphyrin structure did not improve the electrochemical oxidative activity of the sample. These results can be related to the higher surface area provided by the mesoporous silica, as well as to a higher density of defects (better conductivity, which may result in an enhancement of charge transfer processes).

### 3.3. Photoelectrochemical Properties

Interestingly, all the V-porphyrin-based samples, either bulk or supported, present an appreciable photoelectrocatalytic response, although the performance depends on the characteristics of the sample. Figure 9 shows the potentiodynamic polarization curves under dark–light conditions used to make the photoelectrocatalytic characterization of the samples. In this case, by way of illustration, curves for the VTPP and 20VP@SBA catalysts are presented. It can be observed that, upon illumination, both samples were reactive, which is indicated by the different current density pulses observed in the plots. However, the SBA-15-supported porphyrin (20VP@SBA) provided dark current densities (the baseline) significantly lower than the VTPP sample (i.e., the former sample is not undergoing any degradation process in the electrolyte, and there are not parallel electrochemical processes taking place in the system), as well as a much more stable signal throughout the potential range.

The values of dark current densities (*i_dark_*), photocurrent densities (*i_ph_*), and net current densities (Δ*i*), which indicate the relative increase in current density when illuminating the electrodes, are presented in Table 2 for all the samples, at three different potentials. It can be seen that, in general, porphyrins with silica (amorphous and mesoporous) provided appreciably higher net current densities, being that the samples with 20VP@SBA (with and without niobium) were the ones with the best photoelectrocatalytic response. These results indicate that, once optimized, porphyrins supported in silica could be used as efficient photoelectrocatalysts.

## 4. Conclusions

In this paper, the incorporation of V-porphyrins in mesoporous siliceous materials (SBA-15 and Nb-containing SBA-15) has been studied by the insertion of 5,10,15,20-Tetraphenyl-21*H*,23*H*-porphine vanadium(IV) oxide (VTPP) into the hexagonal channels of mesoporous materials (SBA-15). These samples have been compared against the conventional incorporation of VTPP on the surface of an amorphous SiO_2_ carried out by impregnation. In this way, we have optimized the synthesis procedure for preparing siliceous mesoporous materials (SBA-15 type) in which the nature of the V-porphyrin remains unaltered. The characterization of materials by XRD, N_2_-adsorption, FTIR, DR-UV-vis, and EPR confirms the optimal preparation of VTPP-containing siliceous catalysts. However, this has not been completely effective when using a Nb-containing SBA-15 material. In addition, the characterization results suggest that the encapsulation of the vanadyl porphyrin through the mesoporous SBA-15 is achieved effectively and that this encapsulation occurs without significant disruption in supramolecular organization. In fact, FTIR, DR-UV-vis, and EPR spectra indicate no changes related to VTPP molecules when comparing pure VTPP and VTPP-containing siliceous catalysts.

Electrochemical results showed that supporting the porphyrin in silica improved the electrical conductivity of samples, especially with SBA-15, since a notable enhancement in electrochemical performance was observed for this sample compared to both the pure and amorphous silica-supported VTPP ones. This result has been ascribed to the higher surface area of SBA compared with amorphous silica (nanopowder). Moreover, all samples presented n-type semiconductivity and provided an interesting photoelectrochemical response upon illumination, especially silica-supported porphyrins. This is the first time that a photoelectrochemical study has been undertaken using V-porphyrin-derived materials. Therefore, these results could open new opportunities to use these types of materials in electro- and photoelectrocatalytic applications.

## Figures and Tables

**Figure 1 materials-15-07473-f001:**
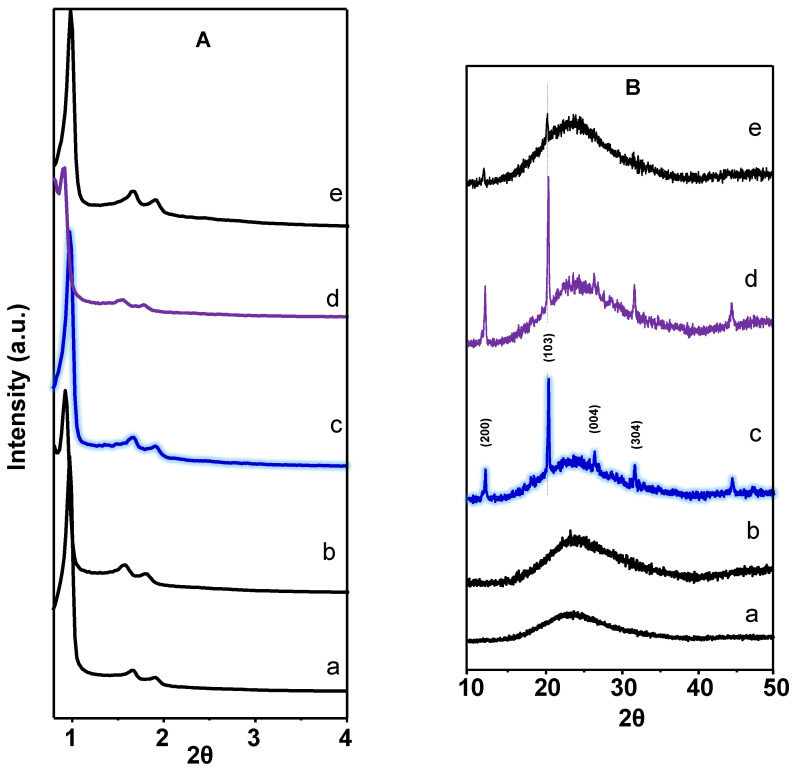
XRD patterns at low angles (**A**) and from 10 to 50° (**B**) of supports and catalysts: (a) SBA; (b) SBA(Nb); (c) 20VP@SBA; (d) 20VP@SBA-Nb; (e) 5VP@SBA.

**Figure 2 materials-15-07473-f002:**
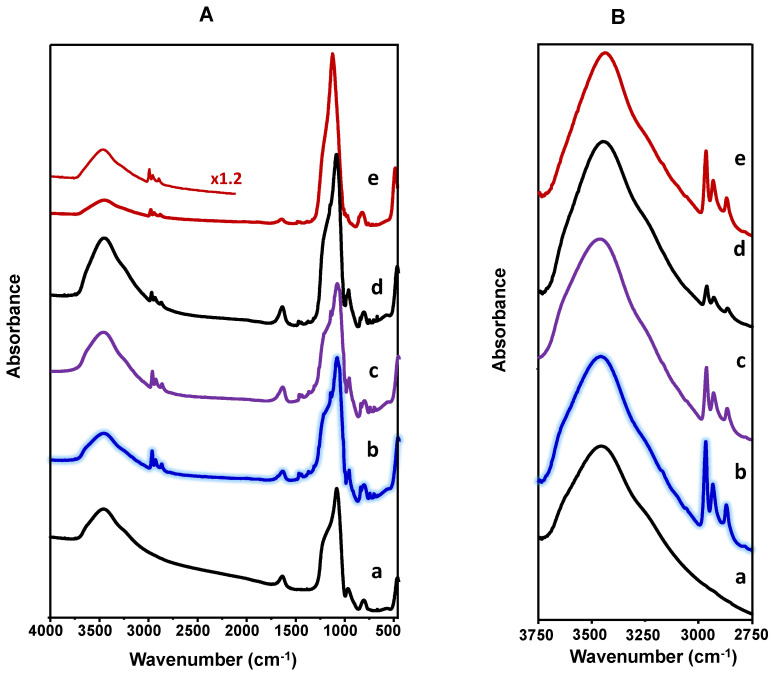
IR spectra of supports and catalysts: (a) SBA; (b) 20VP@SBA; (c) 20VP@SBA(Nb); (d) 5VP@SBA; (e) 20VTPP/SiO_2_. Characteristics of catalysts in Table 1. (**A**) Spectra from 4000 to 400 cm^−1^, (**B**) Spectra from 3750 to 2750 cm^−1^.

**Figure 3 materials-15-07473-f003:**
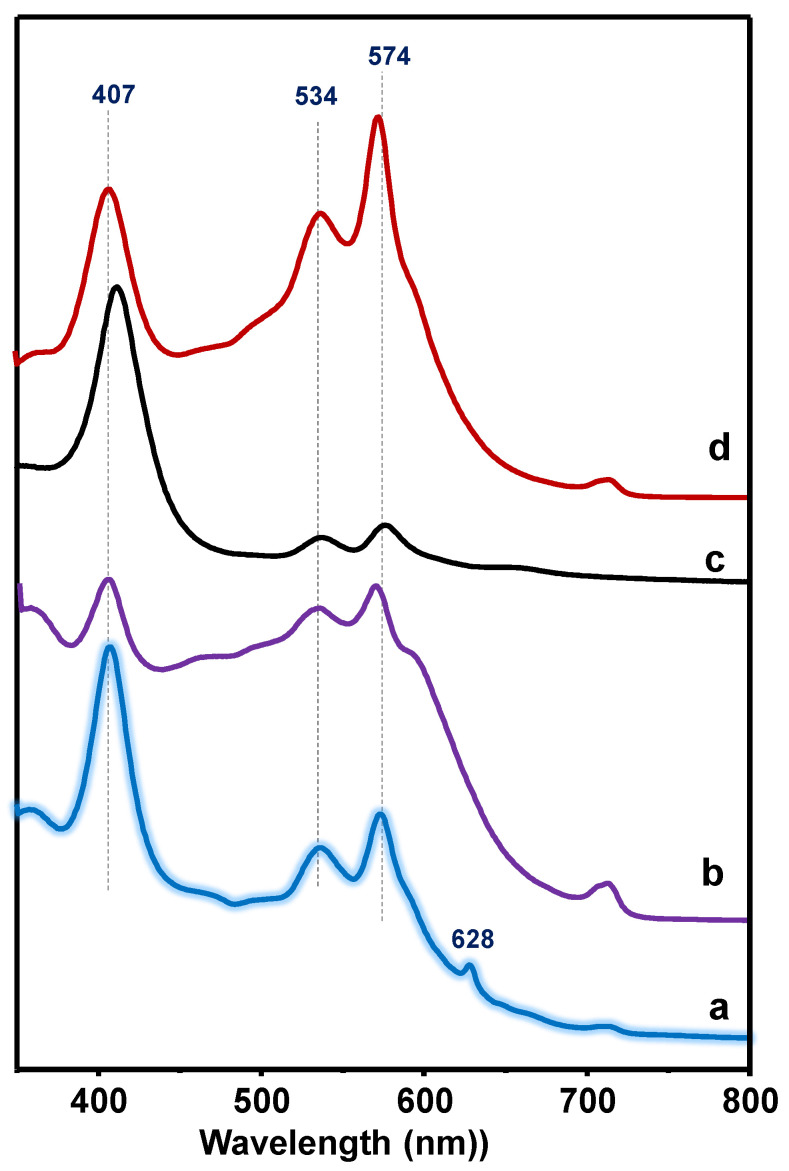
Diffuse reflectance (DR-UV-vis) spectra of catalysts: (a) 20VP@SBA; (b) 20VTTP@SBA-Nb; (c) 5VP@SBA; (d) 20VTPP/SiO_2_.

**Figure 4 materials-15-07473-f004:**
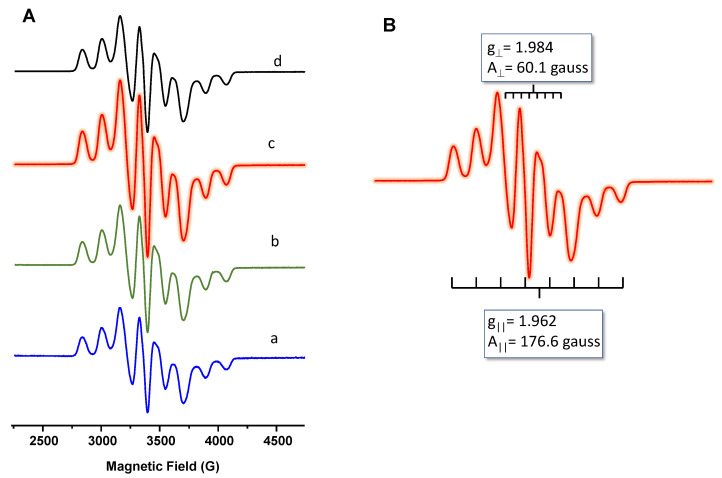
(**A**) EPR spectra of VTPP-containing catalysts: (a) 5VP@SBA; (b) 20VP@SBA-Nb; (c) 20VP@SBA; (d) 20VP/SiO_2_. (**B**) Detailed spectrum of 20VP@SBA catalyst.

**Figure 5 materials-15-07473-f005:**
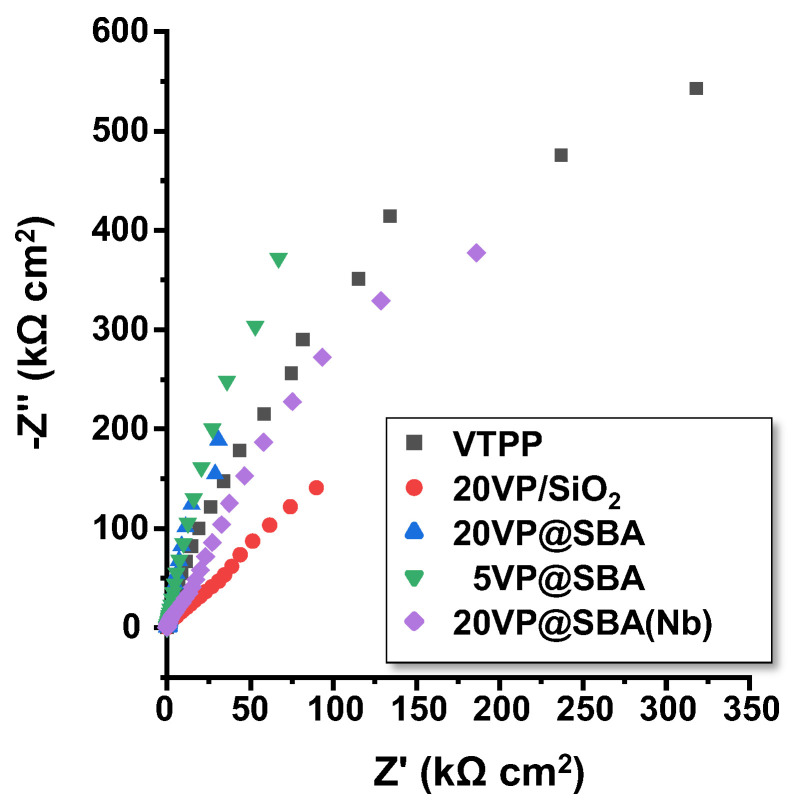
Nyquist plot for VTPP-containing catalysts for the different VTTP-containing catalysts.

**Figure 6 materials-15-07473-f006:**
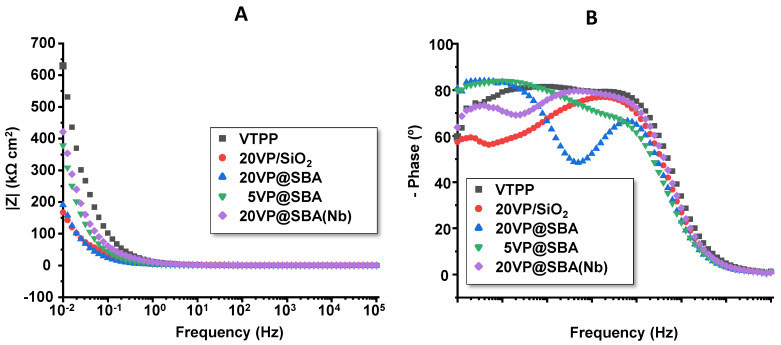
Bode-module (**A**) and Bode-phase (**B**) plots for VTPP-containing catalysts for the different VTTP-containing catalysts.

**Figure 7 materials-15-07473-f007:**
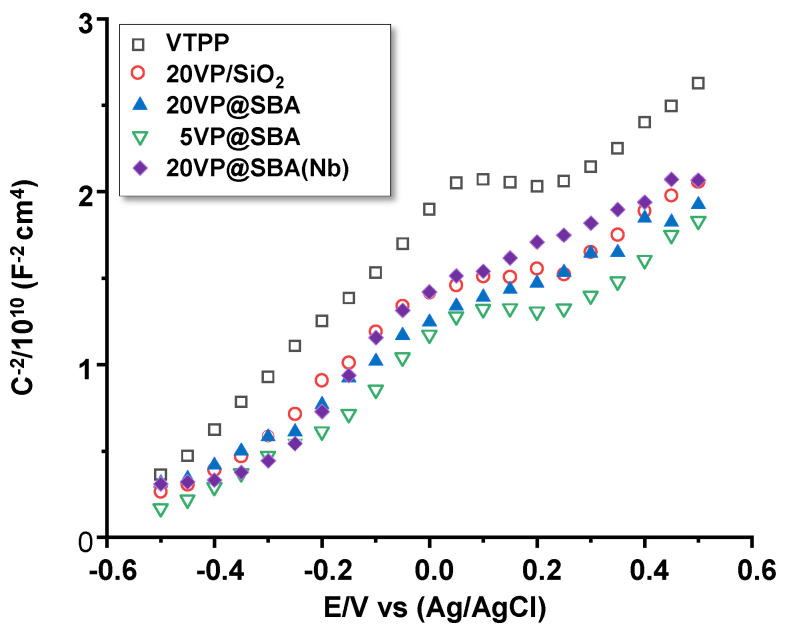
MS plots for the different VTTP-containing catalysts. Characteristics of catalysts in Table 1.

**Figure 8 materials-15-07473-f008:**
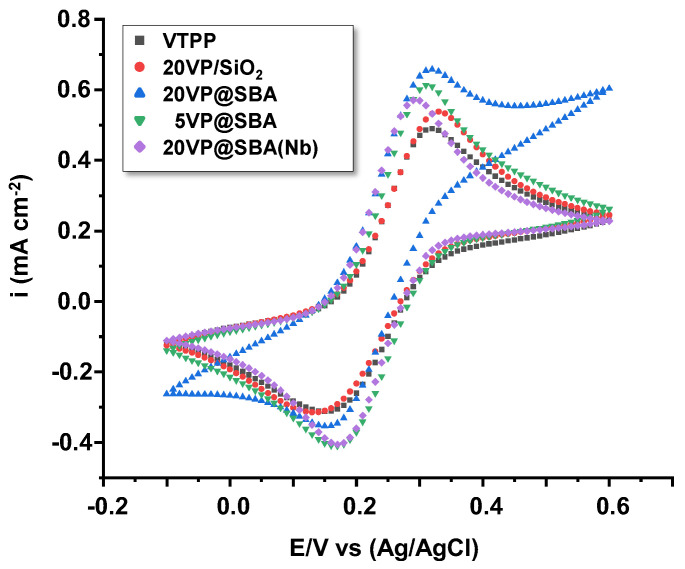
Cyclic voltammetries for the VTTP-containing catalysts (scan rate: 10 mV/s).

**Figure 9 materials-15-07473-f009:**
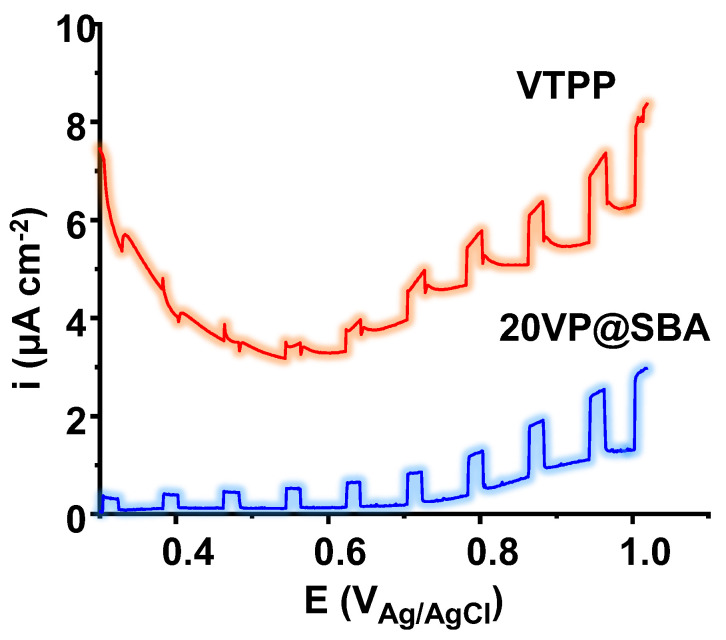
Photocurrent densities vs. potential plots for pure (VTPP) and encapsulated (i.e., 20VP@SBA) catalysts.

**Table 1 materials-15-07473-t001:** Characteristics of the supports and samples synthesized.

Sample ^1^	Support	VTPP Content (wt%)	S_BET_ (m^2^ g^−1^)	Amount of V by EPR ^3^
SiO_2_	SiO_2_	0	200	0
SBA	SBA-15	0	740	0
Nb-SBA	Nb-SBA-15	0	705	0
VTPP	-	100	nd	15.5
20VP@SBA	SBA-15	20	412	2.34
5VP/@SBA	SBA-15	5	520	0.61
20VP@SBA(Nb) ^2^	SBA-15(Nb)	20	350	1.72
20VP/SiO_2_	SiO_2_	20	162	2.51
5VP/SiO_2_	SiO_2_	5	190	n.d.

^1^ VTPP = 5,10,15,20-tetraphenyl-21*H*,23*H*-porphine vanadium(IV) oxide; ^2^ Nb-containing SBA-15; ^3^ detected as V(IV) by EPR.

**Table 2 materials-15-07473-t002:** Dark current densities, photocurrent densities, and net current densities at different potential values for the V-porphyrin-containing catalysts.

Potential (V_Ag/AgCl_)	Catalyst	*i_dark_* (μA·cm^−2^)	*i_ph_* (μA·cm^−2^)	Δ*i* (μA·cm^−2^)
0.40	VTPP	3.50	3.50	0.00
	20VP/SiO_2_	0.30	0.78	0.48
	20VP@SBA	0.10	0.39	0.29
	5VP@SBA	0.05	0.45	0.40
	20VP@SBA(Nb)	0.25	0.70	0.45
0.55	VTPP	3.25	3.45	0.2
	20VP/SiO_2_	0.54	1.12	0.58
	20VP@SBA	0.10	0.50	0.40
	5VP@SBA	0.38	0.68	0.30
	20VP@SBA(Nb)	0.38	0.98	0.60
0.70	VTPP	4.60	4.90	0.30
	20VP/SiO_2_	1.25	2.15	0.90
	20VP@SBA	0.25	0.85	0.60
	5VP@SBA	0.40	1.00	0.60
	20VP@SBA(Nb)	1.15	2.10	0.95

## Data Availability

Not applicable.

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
