# Peer review of "V-Porphyrins Encapsulated or Supported on Siliceous Materials: Synthesis, Characterization, and Photoelectrochemical Properties"

_materials, 2022, doi:10.3390/ma15217473_

Round 1

Reviewer 1 Report

Reviewer comments on Manuscript “V-porphyrins encapsulated or supported on siliceous materials: synthesis, characterization and photoelectrochemical properties”. 

Authors have presented V-porphyrin based catalysts that were tested in photoelectrochemical applications. Impedance measurements confirmed that supporting the porphyrin in silica improved the electrical conductivity of samples, and when using mesoporous silica, current densities associated with oxidation/reduction processes appreciably increased, implying an enhancement of charge transfer processes and, therefore, in electrochemical performance. 

Authors have presented and extensive investigation of the obtained samples by electrochemical and surface analysis methods. Furthermore, their description and results discussion is sufficient. The manuscript quality is quite good, and I’d recommend this manuscript for acceptance after minor revision: 

  • Photoelectrochemical applications should be more extensively presented in introduction part. What do other types of compounds are usually used for such applications? 

  • CV results description should be improved, as authors compare current density values of the samples. Current density depends on the concentration of the sample used for CV measurement and should not be taken into account for interpretation. Ionization potential values or oxidation potential values should be compared and discussed.

Author Response

#Reviewer 1

  • Photoelectrochemical applications should be more extensively presented in introduction part. What do other types of compounds are usually used for such applications?

According to the reviewer‘s comment, a paragraph explaining the basic concepts of photoelectrochemistry, its main applications and the materials usually employed as photoelectrodes, has been included in the introduction section (page 2, lines 53-61). Moreover, references [39-44] have been included.

"Photoelectrochemistry studies, in general, processes taking place at the interface between a semiconducting electrode and an electrolyte where, under the influence of illumination with suitable energy, electron-hole pairs are generated. Photoelectrochemistry has nu-merous attractive applications [39-44], such as the split of a water molecule to obtain its fundamental constituents (H2 and O2), the advanced oxidation of persistent and toxic or-ganic pollutants in water treatment processes, the photelectrochemical reduction of greenhouse gases, such as CO2, or the fabrication of dye-sensitized solar cells (DSSC). Semiconductor metal oxides, such as TiO2, WO3 or different copper oxides, are increas-ingly studied as photoelectrodes, due to their different characteristics.“

  • CV results description should be improved, as authors compare current density values of the samples. Current density depends on the concentration of the sample used for CV measurement and should not be taken into account for interpretation. Ionization potential values or oxidation potential values should be compared and discussed.

According to the reviewer‘s suggestion, the presentation and discussion of cyclic voltammetries has been improved. Current densities, especially the anodic ones, which are higher than the cathodic, has been presented in the text and a more complete comparison between the different samples has been made. Besides, the influence of porphyrin concentration has also been explained, as well as the presence of Nb in the SBA-15 structure. These changes have been implemented on page 11, lines 329-338.

"Indeed, current density values for the VTPP sample were -0.3 mA/cm2 and 0.49 mA/cm2 (cathodic and anodic peaks, respectively), while for the 20VP/SiO2 these values were -0.31 mA/cm2 and 0.54 mA/cm2. When mesoporous silica was used, current densities notoriously increased, especially the oxidation current density for the 20VP@SBA samples, reaching 0.65 mA/cm2. With lower amounts of porphyrin (5VP@SBA), the anodic current density was slightly lower (0.61 mA/cm2) than for 20VP@SBA, indicating lower electrochemical activity towards oxidation processes. With the addition of Nb (20VP@SBA(Nb)), the anodic current density decreased to  0.57 mA/cm-2, so, in this case, the presence of Nb in the porphyrin structure did not improve the electrochemical oxidative activity of the sample. These results….”

Reviewer 2 Report

The authors reported a systematical investigation based on V-porphyrins supported on siliceous materials (VTPP@SBA). The overall looks well, but several issues should be corrected. 

1) Introduction part: The motivation for V-containing materials is not clear. Please add more demonstrations. 

2) The below refs may be useful in the porphyrin-based materials, as follows:

a. Journal of the American Chemical Society, 2008, 130(40): 13214-13215.

b. Journal of Alloys and Compounds (2022): 167189.

3) Provide the crystal parameters in Figure 1. a-b. 

4) Wavenumber index info on Figure 2. 

5) Please provide EIS simulated circuit in Figure 5. 

6) Please note the scan rate info in the caption of Figure 8. 

7) Figure 7. what are the MS plots? the concept is not clear. 

Author Response

The authors reported a systematical investigation based on V-porphyrins supported on siliceous materials (VTPP@SBA). The overall looks well, but several issues should be corrected.

1) Introduction part: The motivation for V-containing materials is not clear. Please add more demonstrations.

According to the comment of the referee, we have added a paragraph indicating the motivation for the study of the V-porphyrins. See changes in page 1:

“The oil currently extracted is, on average, heavier than that obtained several decades ago. This oil, apart from presenting larger hydrocarbons, typically contains larger concentrations of metals, sulfur, oxygen and nitrogen. Thus, V is one of the most abundant metals in the oil and commonly is present in porphyrin structures. Therefore, the search for new applications of these V-containing petroporphyrins is highly interesting. Vanadium (IV)-porphyrins containing materials are also paramount.”

And in page 2:

“The role of vanadium is important in these applications. Then, it has been reported that vanadium in other surroundings as in V-doped nickel sulfide nanoflowers, exhibit notable electrochemical performance [33].”

2) The below refs may be useful in the porphyrin-based materials, as follows:

  1. Journal of the American Chemical Society, 2008, 130(40): 13214-13215.
  2. Journal of Alloys and Compounds (2022): 167189.

According to the comment of the reviewer we have included the second reference suggested, i.e. that in J. Alloys and Comp. Now, this reference 33. On the other hand, we have not added the first reference as we think the relationship with the topic of the article is not strong enough.

3) Provide the crystal parameters in Figure 1. a-b.

Crystal parameters of VTPP compound is included in modified manuscript (see changes in page 5):

“the XRD patterns show several reflections at 2θ= 8.80; 12.28; 20.56; 31.83; 44.54°, which indicate the presence of VTPP with a polycrystalline structure [47, 48], as observed for pure VTPP (see Fig. S4, pattern a), related to a tetragonal phase structure with space P43 [48].”

In addition the main Miller indices related to VTPP, has been included in both Fig. 1B and Fig. S4.

4) Wavenumber index info on Figure 2.

We have modified the manuscript, describing the main IR band of VTPP (page 6):

“The presence of C–H groups was indicated by the bands at 2869 (CH3 asymmetric stretching), 2930 (symmetrical CH2) and 2967 cm−1 (asymmetrical CH2), in addition to those at 1443, 1373 and 705 cm−1 [14, 25, 49, 50], whereas the presence of C–C groups was indicated by the band at 1465 cm−1 [50]. On the other hand, the presence of a pyrrole C–N bond was indicated by the band at 775 cm-1 [50] and bands at 1058 and 844 are related to N-H vibration [49].  All these band are also…”

In addition the main of these band can be seen in Figure S6C, spectrum d.

5) Please provide EIS simulated circuit in Figure 5. 

In this work we have performed EIS measurements to compare the shape of both Nyquist and Bode plots between the different samples and to obtain conclusions from a qualitative point of view. Therefore, we have not used any equivalent circuit to fit the experimental results, since doing a complete EIS study of the samples was not the purpose of the present work. Nevertheless, we thank the reviewer for the kind and constructive observation and, in future works with porphyrin samples, we will perform this analysis in depth.

6) Please note the scan rate info in the caption of Figure 8. 

According to the reviewer’s comment, the scan rate (10 mV/s) has been included in the caption of Figure 8.

7) Figure 7. what are the MS plots? the concept is not clear. 

According to the reviewer’s observation, an explanation about Mott-Schottky (MS) analysis has been included in the “Results and Discussion” section (page 10, lines 300-321).

“In order to investigate whether the different porphyrin samples presented semiconducting behaviour, the capacitance of the interface between the samples (electrodes) and the electrolyte was measured as a function of the applied potential. Mott-Schottky (MS) analysis was performed by plotting the reciprocal square interfacial capacitance versus the applied potential. Figure 7 shows the MS plots for the different samples. First of all, it can be observed that a positive slope was obtained in all plots, according to the following equation for n-type semiconductors:

where C is the measured interfacial capacitance, CH is the capacitance of the Helmholtz layer, εr is the relative permittivity or dielectric constant of the sample, ε0 is the vacuum permittivity (8.85·10-14 F/cm), e is the electron charge (1.60·10-19 C), ND is the density of defects (donor species for n-type semiconductors), E is the applied potential, EFB is the flat-band potential, k is the Boltzmann constant (1.38·10-23 J/K) and T is the absolute temperature.

Hence, for an n-type semiconductor, the plot 1/C2 vs E yields a linear region with positive slope, whose value is inversely proportional to the density of donor species (ND). From Figure 7 it is clear, therefore, that all the samples presented n-type semiconducting behaviour. The value of the slope was similar for all the samples, although for the VTPP the slope was somewhat higher, indicating a lower density of defects. Samples with the vanadium porphyrin and dispersed in mesoporous silica (20VP@SBA and 5VP@SBA) showed slightly lower slope values, meaning that these samples had a higher donor density, although the values depend on the dielectric constant values of the different samples.”

Round 2

Reviewer 2 Report

The revised manuscript looks better. It can be accepted on Energies directly. Congrats!